# Towards More Proactive Sustainable Human Resource Management Practices? A Study on Stress Due to the ICT-Mediated Integration of Work and Private Life

**Kristina Palm [1,2,3,\*], Ann Bergman [†] and Calle Rosengren [4]** 

[1]  Department of Learning, Informatics, Management and Ethics, Karolinska Institute, S-171 77 Solna, Sweden
[2]  Department of Sustainable Production Development, KTH Royal Institute of Technology, S-151 81 Södertälje, Sweden
[3]  Department of Work Science, Karlstad University, S-651 88 Karlstad, Sweden
[4]  Center for Design Studies, Lund University, S-221 00 Lund, Sweden; calle.rosengren@design.lth.se
\*  Correspondence: kristina.palm@ki.se; Tel.: +46-734-607-472
†  Deceased.

**Abstract:** This article discusses sustainability in Human Resource Management (HRM) in the blurred digital working life, focusing on the emotion of stress. Its empirical basis is an activity and emotion diary study conducted with 26 employees of three industrial companies in Sweden. Our results show that work and private life are integrated by digital activities and also by emotions. Due to the extensive use of digital devices, stress in the working sphere is not only connected with work, and stress in the private sphere is not only connected with private life. The study also shows that stress is often episodic and can end due to activities connected with both the trigger and non-trigger spheres. From a social sustainability perspective, this study suggests that HRM should gently extend employee consideration beyond the traditional temporal and spatial boundaries of work, i.e., also including private life when understanding work in the digital age.

**Keywords:** diary; emotion; ICT; stress; sustainable HRM; work–life integration

## 1. Introduction and Aim

The increased use of digital technology has fundamentally affected the conditions framing the organization of working life [1–4]. The digitalization of working life has, among other things, challenged work's traditional boundaries and allowed work to be spatially and temporally "disconnected" [5–8]. Information and communication technologies (ICT) in general, and the smartphone in particular, have enabled employees to stay connected to their work from any place and at any time [9–11]. Such technologies have also contributed to new ways of combining work and other aspects of life [12]. Today, employees can have access to their private lives from their workplaces to a greater extent than ever before. This possibility of combining work and non-work responsibilities is reported by employees as one major advantage of the increased flexibility facilitated by ICT [13–16]. However, research on smartphone and ICT use has also consistently shown that the boundaries between work and family life have become permeable [17], which may increase the risk of experiencing imbalance between work and family life [18]. When work is done at the expense of involvement in private life, there is often a sense of stress and of role conflict [19,20]. On the other hand, more recent research shows that this does not necessarily lead to experiences of imbalance [16]. The digitalization of working life both threatens and provides opportunities regarding employee wellbeing, but it also poses new challenges to organizations in adopting work cultures and management styles to support this new way of working. Kowalski and Swanson for example highlight a trust-based organizational culture

and results-based management systems as central if employees are to manage where and when to work more autonomously [21]. Several researchers have requested that Human Resource Management (HRM) develop and adjust to this new working life [22–24].

Research on stress and recovery in relation to work often only focuses on the work setting (e.g., [25–31]), i.e., on stressors at work and recovery from work. However, given the blurred boundaries between work and private life, the private sphere is often also integrated into the work sphere and sometimes even more than the other way around [12]. It is therefore important to take into account stressors and recovery from the private sphere when trying to understand how HRM practices can be developed to promote a more sustainable working life. The possible integration of these two spheres through ICT brings new challenges, and HRM needs to be studied more in this context.

In recent years, some organizations have moved from a single HRM perspective focused on the gain of the employer towards a dual perspective that also considers the employee's best interests [32,33]. This is sometimes referred to as Sustainable HRM [32–34] and sometimes as Socially Responsible HRM [35].

Sustainable HRM includes the same three core aspects as the sustainable work system model developed by Docherty et al., i.e., economic, social and environmental values. This means that Sustainable HRM has a triple value perspective and not only the traditional focus on economic values for the shareholders [36]. Based on earlier research, Stankevičiūtė and Savanevičienė created five clusters of principles important for the concept of Sustainable HRM: employee competencies, voice of employees, employee–employer relations, care of employees and care of the environment [33]. They showed that Sustainable HRM may reduce work-related stress, work–family conflicts and burnout through a more clearly expressed Sustainable HRM. Furthermore, their study showed that employees' work-related stress would decrease if employees perceive increased care and improved employer–employee relations. Care of employees covers both preservation of employees and flexibility. Preservation of employees includes the views that human resources should be regenerated and not consumed at the workplace, employee health should be preserved (including reducing employee stress), workload should be balanced (necessitating a special focus on work–life balance), and pay should be reasonable. The flexibility perspective focuses on the nature of the work organization, i.e., rotation of employees, substitution of employees and flexible work schedules that allow individuals to match their needs with those of the employer. Employee–employer relations concern employees as equal partners, co-operation between employees, as well as fairness and equal opportunities.

Socially Responsible HRM emanates from the work on Corporate Social Responsibility (CSR) [37]. In comparison with Sustainable HRM, Socially Responsible HRM has a dual perspective, thus focusing on creating shareholder value and on the social aspects of the employees [35]. Dupont et al. [37] focus on recruitment and job access, training and career advancement, and health and wellbeing in the workplace, while Diaz-Carrion et al. [35] also include compensation, performance appraisal and safety issues. Furthermore, Diaz-Carrion et al. highlight the importance of cultural contexts when understanding the work of HRM [35]. They argue that the dominant model of business governance is highly influenced by the country in which the company headquarters is located, and that in turn has a large impact on the HRM practice. In a study comparing European countries, Sweden was a leading nation in the introduction of Socially Responsible HRM [35], probably as a result of a long tradition of strong trade unions in the labor market. In the remainder of this article, we integrate the two strands of HRM presented above into Socially Sustainable HRM, thus highlighting the importance of a social perspective.

In this article, the goal is to discuss the theoretical and practical implications for HRM in a blurred working life. Based on employees' experiences of stress in ICT-mediated work–life integration, our overall purpose is to consider ways of making HRM more proactive and socially sustainable and rendering work more sustainable. Since stress could lead to ill health and sickness absenteeism, i.e., the opposite of a sustainable working life, this article focuses on the emotion of stress, triggers for stress,

the durability of stress and stress breakers. The empirical basis of the discussion is a diary study on Swedish employees' boundary-crossing digital activities and their related emotions.

In Sweden, employers have great responsibility for the health of their employees, as witnessed in a well-defined work environment responsibility framework, including work adaptation and rehabilitation. Recently, this has been expanded to also regulate knowledge requirements, goals, workloads, working hours and victimization [38]. The expansion is due to changes in the labor market and in working life, as well as because of new knowledge about what causes form the basis of work-related ill health in current working life. Managers in the organization are tasked with work adaptation and rehabilitation according to the rules on systematic work environment measures. As regulated in the Swedish Work Environment Act, the Swedish Work Environment Authority's regulations on work adaptation and rehabilitation [39], and the Social Insurance Code [40], employers have to adapt the workplace to the needs of their employees and must take rehabilitation measures—irrespective of the origin of the illness or impaired work ability. Employers are also required to have policies and procedures for handling issues concerning sick leave and rehabilitation. However, according to Prevent [41], a Swedish non-profit organization run by the Confederation of Swedish Enterprise, and the trade union confederations The Swedish Trade Union Confederation (LO) and the council for negotiation and cooperation (PTK), employers cannot passively wait on the Social Insurance Agency's initiatives and measures. These organizations argue instead that, both from staff policy and business perspectives, employers should act at an early stage and take proactive measures to improve employees' work situations before serious illness occurs.

The main contribution of this article is to the research and debate in the area of HRM and sustainable work, and more specifically to the research on stress in relation to work–life integration with ICT. First, since Sustainable HRM suggests a dual perspective emphasizing social sustainability [32–35], this article contributes to the field by starting from employees' day-to-day experiences of stress when discussing HRM policies and practices for a sustainable working life. Second, compared to much research focusing on stress (e.g., [42]) and recovery in work (e.g., [26,28,30,31]), this article takes both the work and the private spheres into account to understand day-to-day stress at work, since the two spheres are often integrated. In addition, we discuss episodic stress [42], the kind of stress that may, if unmanaged, turn into a dangerous extended stress condition. Finally, we contribute to the field of HRM by gently extending employee consideration beyond traditional work boundaries.

## 2. Literature Review

### 2.1. Work–Life Integration

Many theories seek to understand and explain the relationship between work and other life, for instance, work–life balance [43,44], work–life conflict [45,46], or work-to-family conflict and family-to-work conflict [47]. Some research focuses sharply on how one sphere encroaches on the other, such as "work-to-family interference" [48] or work spillover [49], or more positive aspects such as how work engagement enriches employees beyond the contribution of the domain of work [50], while other studies underline the importance of reciprocity [51,52]. Some research shows that digital technology has contributed to work increasingly encroaching on other facets of life, hence the evolution of concepts such as "work extending technology" [4]. Notwithstanding, other studies have shown that digital technology facilitates the reaching of family-related and private matters into working life [7,12,53]. This in turn has led to attempts to find "bridging concepts" such as "boundarylessness" [54], "permeability" [6] or "work–life integration" [55]. The theoretical challenge is to avoid unilateral emphasis on balance or conflict, starting from the premise that influence travels in a certain direction, or regarding boundarylessness as the sole alternative to separate work and family spheres. Kossek [12] found that various strategies for using digital technology to integrate or separate work and life coexist, i.e., there are boundary management strategies that are far more differentiated than the "boundary theory" subdivision into "separators"

and "integrators". Strategies concern the conditions of family and working life, and intentions, motives and claims relating to both work and life.

## 2.2. Stress and Recovery in Working Life

Much research on job stress compares people regarding relatively stable job stressors and fails to take into account the more episodic stress that people experience in their day-to-day lives [42]. Episodic feelings of stress are responses to specific stressful events, may occur either occasionally or often, and are not necessarily enduring aspects of the job [42]. Studies on episodic stress can give greater understanding of the link between stressful workplace conditions (i.e., stressors) and the strain (i.e., psychological and physical detriments to wellbeing) expected to occur within the same workday. The transactional model of stress [56] focuses on episodic stress: "[ … ] the relationship between job stressors and strains is a dynamic process shaped by individuals' specific appraisals of an event as stressful and ongoing attempts to cope with the stressor" [42] (p. 4). Today, when the boundaries between work and other parts of life are permeable [6], it is more likely that there are potentially more and diverse stressors at work and also in the private sphere. It is likely that the total amount of episodic stress increases and therefore the risk of more enduring stress increases too. The body needs recovery during periods of stress [57]. Several researchers state that it is of key importance to have periods of recovery from the stressors one is exposed to at work [28,31,58,59]. Westman and Etzion [60] showed that the wellbeing of employees tends to increase during vacations, but research also shows that the beneficial effect on wellbeing usually is short-lived [61]. Ensuring instant recovery from stressful and demanding working days seems more important [31].

A useful concept in this matter is "psychological detachment", which is defined as the individual's experience of being detached or de-connected from the work situation [62]. Considering recovery from this perspective, there is a need to differentiate between being psychologically or physically absent from work. Psychological detachment from work is much more difficult when work tasks, such as answering emails, are performed outside ordinary working hours and working spaces [27,61,63]. Sonnentag [59] showed that having fewer work-related activities in the evening leads to improved wellbeing by bedtime. A later study by Sonnentag and Bayer [31] showed that individuals with a greater ability to detach themselves from work experienced more positive moods and less strain. Derks et al. [27] found strong evidence that smartphone use disturbs the important process of recovery, especially when work-related activities are engaged in at home. In their qualitative diary study of the service sector, Gombert et al. [64] showed that employees who used their smartphones extensively for work-related tasks in the evening, but at the same time experienced good sleep quality, increased their chances of effectively managing self-control requirements during the next working day. On the other hand, if sleep quality was poor, chances were reduced. Besides psychological detachment, sleep has been proven to be of the utmost importance when recovering from stress (e.g., [63]).

Not only the time itself spent on emailing at all possible times and in all places threatens recovery, but also the experience of being constantly available (c.f. [65]). The experience of never being free from, or mentally and/or digitally disconnected from, work may thus lead to an increased amount of stress, and in the long run to decreased wellbeing [3,25,31,57].

## 3. Materials and Methods

An activity diary [66] was developed to collect data on boundary-crossing digital activities between work and private life and their related emotions. The study was approved by a Swedish Research Ethical Committee (2016/2511-31).

### 3.1. Activity Diary

For seven days, respondents logged digital activities that cross the boundary between work and private life. The diary was divided into seven time slots during a day (and night). The respondents summarized each time slot with their general emotional state(s): "on top of things", stressed, calm,

creative, chaos, engaged, frustrated, bored, confused, energetic and tired. We chose emotional states that are not the result of psychiatric disorders. It was also possible for the respondents to mark "something else" and write down their own feelings, or to comment on the time slot using a free text answer. This article is based on the emotions and free text answers. The diary also provided information on how it should be completed, basic questions (name, age, gender, family constellation and type of employment agreement) and an example entry. The diary was developed through an iterative process in which all three researchers participated.

## 3.2. Context, Participants and Collection of Data

Three large (>250 employees) industrial companies took part in the study: (1) a Swedish site of a global science-led biopharmaceutical business, "the Pharma"; (2) a Swedish site of a global developer and supplier of technologies, automation and services for the pulp, paper and energy industries, "the Tech"; and (3) a site of a Swedish forest and pulp production company, "the Pulp".

The first inclusion criterion was that the participants should be able to locate part of their work outside the workplace and outside normal office hours. For some reason, not all our participants could do so, something that we only learnt after we analyzed their diaries. We regard this as a positive mistake since the analyses show interesting results and are therefore included in the study. We also chose to include a smaller number of blue-collar workers, because the companies were interested in understanding that group. The second to fourth inclusion criteria were that participants should be employees, managers or HR representatives, with an overweight on employees; distributed among women and men; and distributed among four chosen life stages. The life stages are based on chronological (years lived) and social age (social role/function in society, e.g., child, parent and worker) and represent individuals without children, with children of different ages and with elderly parents. The chosen life stages follow the chronological ages of 24–34; 35–44; 45–54; and 55–67.

We asked HR for a longer list than we needed of people who fulfilled the above inclusion criteria, and from that list individuals were chosen for contact. In this way, HR and the employer did not know who participated. Individual 10-min meetings were booked with the participants. During these meetings (held at their workplace), we went through the diary and how to complete it. Participants were asked to send/hand the diary back to the researcher as soon as they could.

## 3.3. Processing and Analyzing the Data

The diaries were copied, and the originals were delivered back to the participants. After the diaries had been copied, the emotions were entered in a specific color in a scheme for each time slot on each day presented, and when comments were written these were also put in the scheme, making it possible to connect emotions with special comments.

A content analysis was performed on the emotion of stress. We analyzed comments regarding stress, often with an explanation of why participants were stressed or stopped being stressed, and we also analyzed the sphere to which the emotion of stress belonged and the sphere in which it emerged. All diaries were analyzed, but seven did not contain any emotions or connected activities and have therefore been excluded from the analyses of this article. In the final analysis, 24 participants were included: 12 from the Pharma, 7 from the Tech and 5 from the Pulp, see Table 1.

**Table 1.** List of participants.

| Characteristics | The Pharma | The Tech | The Pulp |
|---|---|---|---|
| Total no. | 12 * | 7 | 5 |
| Gender | | | |
| Men | 6 | 1 | 2 |
| Women | 6 | 6 | 3 |
| Age | | | |
| 24–34 | 2 | 1 | 2 |
| 35–44 | 0 | 2 | 2 |
| 45–54 | 8 | 4 | 1 |
| 55–67 | 1 | 0 | 0 |
| Child/ren at home | 8 | 5 | 4 |
| Hierarchical level | | | |
| Managerial role | 1 | 1 | 2 |
| Employee | 11 | 5 | 3 |

* One with unknown age.

## 4. Results

The diaries did not capture stress that spanned time and space without interruption, perhaps because people who experience continuous stress chose not to participate. Instead, the analysis suggests that the participants experienced episodic stress that occurs on separate occasions, originating from a variety of factors linked to work or private life, and that is replaced by other feelings after a fairly short time. Sometimes the stress was boundary-crossing and sometimes not.

### 4.1. Private Stressors that Emerge in the Work Sphere

Everyday family affairs, such as misunderstandings about a child's medical appointment that were untangled via mobile phone or a child calling in the middle of a meeting, did create stress in the work sphere. Other stressors were extraordinary events such as a house sale or booking accommodation for a private yearly sports event, which were managed at work through text messages on the phone or booking sites on the computer:

> *Monday, 9–12 am, stressed and engaged: "Trying to book accommodation for this year's major sports event."* (The Pharma, male, 52)

Private stressors that had cross-boundary effects in the sphere of work were not always unambiguously digital, but sometimes also analog, such as not being able to get away from work at a certain time when a friend or family member was waiting or having to leave the workplace even if one does not feel ready:

> *Friday, 3–5 pm, stressed and frustrated: "Hassle with an analysis and my partner is waiting grouchily outside."* (The Pharma, female, 48)

However, digital technology was the communicative link between the two spheres, for example by sending an SMS to say that one is running late.

### 4.2. Work-Related Stressors that Emerge in the Private Sphere

Most respondents were satisfied with their cross-boundary digital activities, but the content of the activity or the time spent could create experiences of stress. When colleagues sent emails late one evening on a public holiday, for example, it was experienced as stressful, especially when the content worried recipients, such as bad news:

*Wednesday, 11 pm–7 am (night), stressed: "Read emails late in the evening and became a little stressed by one email saying that I maybe missed out an important thing. I slept badly since I thought about it during the night (it later turned out that I did not miss it)."* (The Tech, female, 42)

Some respondents experienced the need to catch up in the evenings and weekends as well as to plan the next day's work the night before as stressful, or as described below, found it stressful to wake up early in the morning to start working:

*Wednesday, 11 p.m.–7 a.m. (night), on top of things, stressed, tired: "Woke up at 4 already and could not get back to sleep. Started to work at 4:04 a.m. Went to work at 6:27 a.m."* (The Pulp, female, 31)

Others, on the contrary, felt that catching up and planning work led to "being on top of things":

*Sunday, 9–12 a.m., "being on top of things": "Started to check what had to be done before Monday. Activities at work made me delayed with emails and other tasks."* (The Pulp, male, 51)

*4.3. Stress: Duration, Breakers and Recovery*

One of the study's more unexpected results was that the individuals' emotional states changed considerably over a day. Both positive and negative feelings were short-lived and changing. Based on this result, we analyzed factors contributing to the changes related to stress. Whether stressors emerged from work or from private life, three categories of stress breakers were distinguished.

(1)  Stress broken by social relations, such as dinner with a friend or an uplifting conversation with a colleague.

  *Thursday, 9–12 a.m., stressed and frustrated: "Problems accumulate when you don't have time for all emails before all meetings. The report still doesn't work!"*

  *Thursday, 12–3 p.m., calm: "Feeling the calmness return after a re-energizing lunch with a former colleague."* (The Tech, female, 53)

(2)  Stress broken by digital connectivity that provided urgent knowledge, for example being told that everything was okay via email.

  *Friday, 3–5 p.m., stressed and frustrated: "Hassle with an analysis and my partner is waiting grouchily outside."*

  *Friday, 5–8 p.m., on top of things and tired: "Couldn't stop worrying whether everything went well, so I logged in and checked that the analysis went well. Nice, now I can let the weekend begin."* (The Pharma, female, 48)

(3)  Stress broken by completing activities, such as completing a course that was experienced as stressful beforehand or submitting a visa application in time.

  *Wednesday, 3–5 p.m., stressed: "Continuing work with the visa application that needs to be posted at the latest at 6 pm today."*

  *Wednesday, 5–8 p.m., stressed -> calm: "I posted my visa application at the last minute."* (The Pharma, female, 27)

(4)  In addition, a fourth stress breaker specifically concerns actively disconnecting from work. Stress was broken by the advent of the weekend as a symbolic marker, such as closing the work email and welcoming Friday night rituals such as cooking and socializing with family. This allowed people to start relaxing and stop thinking about their work:

*Friday, 12–3 p.m., on top of things, engaged and stressed: "On top of things, but at the same time stressed since I have to compile the week's visits at another unit. At the same time, you have to keep the everyday work running. People who need answers and cannot wait."*

*Friday, 3–5 p.m., on top of things, engaged and tired: "Email ends for today, pick up kids."*

*Friday, 5–8 p.m., on top of things and engaged: "Supper, weekend, unwind, nice with Friday so you can unwind and aren't focused on work."* (The Tech, female, 47)

## 5. Discussion

The goal of this article was to study stress in the digital working life and discuss the theoretical and practical implications for HRM in such a blurred working life. Several researchers have called for an HRM practice that takes these permeable boundaries into account to develop a more sustainable working life [22–24]. Specifically, our findings indicate that that work and private life are not only integrated through activities, but also through emotions, e.g., stress. By pointing out that stress at work not only emanates from work, but also from the non-work sphere, we contribute to the discussion on how private life could be incorporated into HRM thinking. To date, the HRM literature has focused on the workplace context, and there is a lack of theory and research about the effects of private life ICT-mediated activities on HRM.

A strong finding of this study is that stress is experienced when private activities are carried out in the work sphere and vice versa. In line with the transactional model of stress [56] and studies by Pindek et al. [42], our study confirms that everyday stress is often episodic. Another contribution of the study is that the spatial and temporal boundaries between work and private life do not prevent stress from emerging in one sphere while emanating from the other. Furthermore, stress can be broken, thus enabling recovery, through engaging in activities from both the "trigger sphere" and the "non-trigger sphere". As reflected upon in the theoretical frame, it is likely that there are more stressors in total in both spheres when work and private life are integrated by ICT. This could lead to fewer opportunities to recover from stress. Most of the recovery studies regarding work focus on the importance of detachment from work (e.g., [25–30]), but when these two spheres are integrated, it might be as important to focus on how individuals could detach from possible strain from both spheres. Maybe individuals need recovery spaces that provide physical, digital and mental detachment in order to be psychologically detached from both spheres. Studies show that there are strong links between the organization of work, specific characteristics of different professions, and conditions for availability for work and for family [65,67]. Other research shows that some organizations take the merging of work and family into account and facilitate this merger for their employees [68]. Research indicates that organizations' adoption of family-friendly policies and working hour arrangements leads to greater job satisfaction, improved performance, reduced sick leave and a better work–life balance among employees [47,69–71]. One could argue that these organizations have adopted a more Socially Sustainable HRM in line with Stankevičiūtė and Savanevičienė [33]. They have taken a small step towards integrating the whole life picture of the employee. At the same time, some research suggests that there is a mismatch between what employers consider to promote work–life balance and employees' perceptions [72]; that a large part of the work–life balance discussion is conditioned by employers' interests; and that flexible working conditions also tend to generate conflicts between home and family [5,25,73,74]. Research further indicates that organizations and managements prize employees who have "multi-availability", and this tends to reduce problems to individual rather than organizational propensities [75,76]. Earlier research thus suggests that HR policies that enable individuals to combine work and private life through flexible work arrangements could create a socially sustainable working life [33], and in general more consideration needs to be given to the employee. When formulating these policies, HRM needs to have a dual perspective where the employee's best interests and listening to the voice of employees are also included [33]. One important aspect of breaking stress patterns found in our empirical results is that people must manage stressors in order to release the stress and enable recovery. HRM could, e.g., introduce peer-support groups for reflecting

and talking about stress, both on an individual and organizational level (e.g., [77]), helping people to find strategies to finish whatever causes stress. More generally, HRM and managers could support individuals by talking about stressors, good ways of resolving stressful issues, and detaching and recovering from causes of stress.

When employees are on long-term sick leave for fatigue syndrome, for example, Swedish employers have extensive responsibilities in order to help them return to work [39]. This often means that employers must understand the bigger causal picture of the illness and the needs of the individual, which also includes understanding the private sphere. Research has, e.g., shown that the quality of one's marriage could be of great importance to the risk of cardiovascular disease, and a combination of marital conflict and job strain was a particularly strong predictor of atherosclerosis progression [78]. In practice, understanding the whole life picture is usually one part of the rehabilitation effort from management and is seldom questioned by employees or trade unions. However, if employers were to ask questions about the private sphere to promote health proactively, this would be met with trade union skepticism. The main argument is that employers should not interfere with what employees do in the private sphere or during their free time. The question of where HRM responsibilities start and end is a delicate one: it tends to trigger different emotions, especially resistance from the trade unions' side and curiosity from HRM practitioners. When employers, employees and unions cannot discuss the fluidity of boundaries between work and life, nor the causes and consequences of such fluidity, the chances of creating a sustainable work–life situation for the employee are reduced.

The main benefit of Socially Responsible HRM [33] is to be more proactive in the health and wellbeing matters of the employee. If Socially Responsible HRM were also to include the private sphere, it might be even more successful in increasing health and wellbeing, for example, in considering the effects of work on private life, and vice versa. In such a case, we suggest a sixth cluster for the Sustainable HRM by Stankevičiūtė and Savanevičienė [33]: employees' whole life situation. This is partly included in the cluster "care of employees" as flexibility in when and where work is done, but adding a new cluster would emphasize the importance of understanding private life. Foremost, this perspective could make not only HRM and managers aware of the whole life situation of their employees, but it could also make individuals more conscious about their whole life situation and this may direct a person to a healthier and more sustainable working life. Thus, this leads to a strategy that is positive for that individual, their family and the employer. Even though we foresee potential positive impacts of extending HRM, there are also risks involved if HRM engages in the individual's private sphere. The employer could for example use information from the employee's private sphere in a way that only favors the employer, or individuals may resign their own responsibility and leave that with the employer.

Although the above discussion might not extend actual employer responsibilities in the work and private continuum, it does move the boundary of consideration and in turn necessitates a changed mindset regarding integrated working life.

## 6. Limitations and Future Research

Although our data were collected during a week and seven times per day, which lends credibility to our conclusions concerning stress in the digital working life, a few limitations suggest the need for careful interpretation. First, in the activity diary only cross-boundary activities were requested, and even though we asked for general emotions and comments on each time zone, it is possible that more cross-boundary activities and feelings were reported than non-boundary-crossing activities. A second limitation may stem from the sample being drawn from only a single country and culture. Sweden is a country where the employer already has a great responsibility for the health of their employees, which may be the reason why HRM practices are already more developed towards sustainability [35]. Third, this research does not study the eventual effects of a broadened HRM; instead, it raises the discussion of whether this could be one way for employers to be more proactive in health issues. Fourth, this research was performed in large private companies, but challenges with

spatially and temporally disconnected work can be found in different kinds of organizations, including small companies and publicly owned organizations. We therefore argue that the results of this study should be of interest for other kinds of organizations than those studied here.

Even if the question of HRM responsibility in the permeable working life is inconvenient or maybe precisely because it is, this issue warrants further discussion. Participatory research (e.g., [79]) could be one way to investigate if and how Sustainable HRM actions could be extended to include the whole life situation. Different activities evidencing more integrated work–life HRM should be tested and evaluated both from the perspectives of employers and employees.

## 7. Conclusions and Practical Implications

In an integrated working life, HRM should consider both the work and private spheres when it comes to stressors, stress breakers and recovery. Based on earlier research, work and family policies should be developed with a dual HRM mindset, i.e., including a socially sustainable perspective [33]. Finally, if HRM were to expand the conception of their purview, i.e., by including the private sphere, more proactive efforts may be taken to facilitate a sustainable working life. This article does not claim to provide any final conclusions or practical guidelines for how HRM can create a more socially sustainable working life. On the contrary, it illuminates issues that should be studied and discussed further, especially in the digitalized working life where the fluidity of work and private life tends to push for more integration.

Our findings also have practical implications. If organizations and management were to care about the employee's private life to a slightly larger extent than they do today, they could positively affect their employees' whole lives. In an earlier stage of our research, we discussed the palette of boundary management strategies between work and private life with managers, HRM and employees, and this proved to be a fruitful way of making people conscious about their own behavior and also that of their colleagues, managers and partners. This awareness can lead to changed behavior or a better understanding of different behaviors, and thus to less frustration and fewer conflicts in a work team. That is, a situation can only change by making the individual aware of it, its causes and its consequences. HRM could for example give employees consciousness-raising tools by addressing both their work and private situations, and, as a result, their overall working life situation. Employees might for instance be able to create more sustainable working lives when consideration is given to the fact that larger private events as well as larger work assignments may need to be adapted to function sustainably in the other sphere.

**Author Contributions:** Conceptualization, K.P., A.B. and C.R.; methodology, K.P., A.B. and C.R.; validation, K.P., A.B. and C.R.; formal analysis, K.P., A.B. and C.R.; investigation, K.P., A.B. and C.R.; resources, K.P., A.B. and C.R.; writing—original draft preparation, K.P.; writing—review and editing, K.P., A.B. and C.R.; project administration, K.P., A.B. and C.R.; funding acquisition, K.P., A.B. and C.R. All authors have read and agreed to the published version of the manuscript.

**Funding:** This research was funded by AFA Insurance, grant number 150483.

**Acknowledgments:** A special thanks to the respondents and the companies participating in this study.

**Conflicts of Interest:** The authors declare no conflict of interest.

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
