# Peer review of "Towards More Proactive Sustainable Human Resource Management Practices? A Study on Stress Due to the ICT-Mediated Integration of Work and Private Life"

_sustainability, doi:10.3390/su12208303_

Round 1

Reviewer 1 Report

Could the authors add a few words about the "boundary theory" subdivision "separators" and "integrators"?

The conclusions from the research are correct. The authors may consider conducting future research with standardized psychological tools.

It should be noted that the study was conducted in Sweden, which is a country with a high HRM standard, and not all the postulates can be implemented in countries with other management cultures.

------------------------

Introduction and aim; Literature review

The authors rightly note that new technologies allow constant contact with work. Unfortunately, this may endanger the well-being of employees. It is a challenge for both employees and their managers. Good management practices are taken into account in Sweden. Sweden was a leading nation in the introduction of Socially Responsible HRM. The blurring of the private and professional spheres is a challenge for modern managers. They must pay more attention to the personal life of their employees. New communication technologies are a key factor here. Taking into account both the sphere of work and the private sphere is important to understand the daily stress at work, as these two spheres are often very closely integrated with each other. The theoretical part is well written. The authors refer to the most important theories and research results.

Materials and Methods

The study was very carefully planned, which is an unquestionable advantage.

Results

As the presented study is of a qualitative type, it is difficult to assess it in the context of other studies.

Discussion

The discussion is understandable and well written. It is based both on the available literature and shows its gaps. the authors indicate that "stress is experienced when private activities are carried out in the work sphere and vice versa." Advice for managers takes into account the results obtained in the context of previous research. On the one hand, the interest in the private life of managers indicates their competences, and on the other hand, it may not be accepted by some employees.

Limitations and Future Research

Limitations have been correctly identified. As the study is qualitative, it is worth considering the use of standardized psychological tests in future research.

Author Response

Thank you for your careful reading of the manuscript and comments. 

We have not added suggestions for future research with standardized psychological tools as this is an area out of our knowledge. There is a risk that that we will suggest something that is not valued as appropriate methods.

Reviewer 2 Report

Referee’s Report on the paper entitled “Towards more proactive sustainable human resource management practices? A study on stress due to the ICT-mediated integration of work and private life”.

Goal: this paper uses qualitative data to assess the role of ICT in the stress emotion experienced by workers. The authors use data collected in three firms in Sweden.

Strengths: the paper is well written and provides interesting results on employees’ ICT usage linked with the issue of their stress emotion. Nevertheless, as I'm not highly familiar with qualitative analysis, I let the Editor and the other referees give their feedback on its quality.

Weaknesses: some parts of the paper need to be revised to improve the paper.

Therefore, I request a revision of the paper that fixes the following issues.

Major remarks:

  1. As the role of HRM practices is not directly examined with the collected data, the authors need to restructure their paper. They should not, indeed, write a dedicated sub-paragraph in their literature review on it (4.3) and they should only discuss this issue in the following sections: 1. Introduction, 5. Discussion, 6. Limitations and future research and 7. Conclusions and practical implications.

  1. The authors need to add a new Table that presents the main socio-economic characteristics of their respondents (age class, gender, hierarchical level, with child, sector of activity, size class of their company, etc.) and the average value of these characteristics for the Swedish labour market in order to offer to the readers statistics that will permit to assess the generalizability or specificity of the studied sample.

  1. In 4.2, after line 284 the authors should add an example sentence from participants regarding the ‘contrary’ behaviour.

  1. Page 9, Line 423: one or two reference(s) of ‘participatory experimental research’ should be added.

Minor points:

  1. There are some typos that need to be fixed, e.g.:
    1. Page 2, Line 88: ‘2019 &2018’ where a space is missing between & and 2018.
    2. Page 4, Line 172: and ‘e’ is missing in ‘Therefor’.

  1. Page 5, Line 204: ‘Ethics.’ should be removed or the authors should create a dedicated paragraph.

  1. Page 9, Line 412: ‘zoon’, is it a typo? or what it means?

  1. Page 10, Line 439: the date of publication of Author (XX) is missing. And Page 11, Line 476, the full reference needs to be included.

  1. Page 13, Line 591: the full reference of Prevent (2019) needs to be included.

Author Response

Thank you for helpful feedback. All major remarks are manged and commented on below. All minor points are managed, but not commented. All major changes and some minor changes are made with track changes in the manuscript.

  1. The text on HRM practices has been moved to the introduction from the literature review. This was followed by some additional changes in the introduction in order to make the text flow in a good manner. 

  1. A table that presents the main socio-economic characteristics (that we have) of the respondents (age, gender, hierarchical level, with child, sector of activity) is included in the method section. The size of the companies is included in the running text. In the methodological concerns we have written about the generalizability  of the studied sample.

  1. In 4.2, after line 284 we have add an example sentence from participants regarding the ‘contrary’ behaviour.

  1. Page 9, Line 423: one reference of ‘participatory experimental research’ has been added.